# Discovering Common miRNA Signatures Underlying Female-Specific Cancers via a Machine Learning Approach Driven by the Cancer Hallmark ERBB

**DOI:** 10.3390/biomedicines10061306

**Published:** 2022-06-02

**Authors:** Katia Pane, Mario Zanfardino, Anna Maria Grimaldi, Gustavo Baldassarre, Marco Salvatore, Mariarosaria Incoronato, Monica Franzese

**Affiliations:** 1IRCCS Synlab SDN, 80143 Naples, Italy; katia.pane@synlab.it (K.P.); annamaria.grimaldi@synlab.it (A.M.G.); direzionescientifica.irccssdn@synlab.it (M.S.); mariarosaria.incoronato@synlab.it (M.I.); monica.franzese@synlab.it (M.F.); 2Molecular Oncology Unit, Centro di Riferimento Oncologico di Aviano (CRO), IRCCS, National Cancer Institute, 33081 Aviano, Italy; gbaldassarre@cro.it

**Keywords:** microRNAs, TCGA, machine learning, female-specific cancers, estrogen-dependent cancer, *ERBB* family, bioinformatics

## Abstract

Big data processing, using omics data integration and machine learning (ML) methods, drive efforts to discover diagnostic and prognostic biomarkers for clinical decision making. Previously, we used the TCGA database for gene expression profiling of breast, ovary, and endometrial cancers, and identified a top-scoring network centered on the *ERBB2* gene, which plays a crucial role in carcinogenesis in the three estrogen-dependent tumors. Here, we focused on microRNA expression signature similarity, asking whether they could target the *ERBB* family. We applied an ML approach on integrated TCGA miRNA profiling of breast, endometrium, and ovarian cancer to identify common miRNA signatures differentiating tumor and normal conditions. Using the ML-based algorithm and the miRTarBase database, we found 205 features and 158 miRNAs targeting *ERBB* isoforms, respectively. By merging the results of both databases and ranking each feature according to the weighted Support Vector Machine model, we prioritized 42 features, with accuracy (0.98), AUC (0.93–95% CI 0.917–0.94), sensitivity (0.85), and specificity (0.99), indicating their diagnostic capability to discriminate between the two conditions. In vitro validations by qRT-PCR experiments, using model and parental cell lines for each tumor type showed that five miRNAs (hsa-mir-323a-3p, hsa-mir-323b-3p, hsa-mir-331-3p, hsa-mir-381-3p, and hsa-mir-1301-3p) had expressed trend concordance between breast, ovarian, and endometrium cancer cell lines compared with normal lines, confirming our in silico predictions. This shows that an integrated computational approach combined with biological knowledge, could identify expression signatures as potential diagnostic biomarkers common to multiple tumors.

## 1. Introduction

Breast Cancer (BC) is the most common cancer in women. Among gynecological tumors, uterine endometrial corpus cancer (UCEC) is the most frequent in developed countries, while ovarian cancer (OV) is the most lethal [1,2]. Early detection of these two gynecological cancers is challenging as effective screening methods are lacking, whilst the heterogeneity of breast cancer strongly impacts its prognostic and clinical outcomes [3,4].

Hence, the use of multidisciplinary approaches to exploit genotype–phenotype relationships could aid diagnosis and treatments by identifying novel, non-invasively detectable biomarkers, and through the discovery of novel molecular targets [5]. Nowadays, omics data integration and machine learning (ML) methods are at the forefront of big data processing techniques to solve clinical problems. Machine learning involves complex algorithms and learning methods to make self-improved decisions for a broad range of applications using large datasets [6,7].

On the other hand, integrated computational approaches and large-scale cancer molecular profiling from The Cancer Genome Atlas (TCGA) has provided a global view of genes that are common oncogenic drivers across distinct tumor types [8,9,10,11]. A notable example is the Erb-B Receptor Tyrosine Kinase family of *ERBB* genes; one of which, isoform *ERBB2*, also commonly referred to as *HER2*, has recently shown clinical relevance not only in breast cancer, but also in gynecological malignancies [9,12,13]. However, the *ERBB* biological networks contributing to estrogen-dependent tumors are still unclear.

Some regulatory non-coding RNAs, such as microRNA (miRNAs), may target the *ERBB* family genes and regulate their activity, as they do for other eukaryotic genes. Many researchers have studied miRNAs and their target associations in distinct cancer types [14,15,16], including gynecologic cancers [17,18,19,20,21]. In female malignancies, a growing number of in vitro evidence has highlighted some unexpectedly expressed miRNAs [17,22,23]. These generally act post-transcriptionally by reducing target mRNA stability or by inhibiting target mRNA translation [24]. These short (18–24 nucleotide long) molecules have great clinical significance, and their diagnostic applications are widely recognized, especially in BC [25], and in ovarian [26] and endometrial cancers [27].

So far, tumor-specific miRNAs from breast, ovarian, and endometrial cancer cell types have proved to have in vitro discriminatory capabilities [22]. Hirschfeld et al. [22], analyzed the expression changes of 25 miRNAs in the three cancer types. They revealed in vitro a subset of BC-associated miRNA (let-7b, miR-21, miR-27a, miR-30a, miR-30c, miR-30e) with distinct expression levels compared with the gynecological tumor types, ovarian and endometrial cancers. Consistent with this finding, miRNA-21 has also been proven to be a common signature across the breast, endometrial, ovarian, cervical, and vulvar cancers by others [17], while miR-145, miR-200b, miR-155, miR-205, miR-34, miR-92, and miR-101, were present in most female malignancies, except for vulval cancers [17]. There is a growing demand for diagnostic and prognostic biomarkers for cancer in women. So far, most associations of specific miRNA types with the disease result from evidence-based findings; however, given the large amount of available miRNA data, there is potentially a vast variety of applications in supervised ML-based models in oncology [7].

Many ML models have been constructed by analyzing miRNA expression data for a single tumor type, such as breast [28], ovarian [29], and endometrial [30], as well as other cancer types [31,32,33]. Interestingly, in ovarian cancer research, a diagnostic model using circulating miRNAs from pooled serum cell lines and clinical samples across various cancer types as biomarkers of ovarian cancer did not allow discrimination between benign, borderline, and malignant ovarian lesions, although one set of miRNAs showed efficacy in differentiating distinct tumor types [34].

Multi-relational data fusion has been applied to miRNAs and pathways to identify biological networks. Recently, Wang et al. found cooperative driver pathways by integrating TCGA miRNA-seq from BRCA, UCEC, and OV projects by weighted adjacency matrices of the heterogeneous network [35]. Moreover, the TCGA consortium analyzed the molecular features of five tumor types (four gynecological tumors plus breast), identifying clinically significant subtypes and suggesting potential therapeutic targets [10]. However, to the best of our knowledge, an integrated computational pipeline and an ML model for evaluating commonly deregulated miRNAs in BC, OV, and UCEC cancers have not yet been exploited.

Combining these modern processing techniques might yield computer-aided models for predicting common patterns of miRNAs. Our previous integrated computational analyses discovered a leading protein–protein interaction (PPI) network underlying breast, endometrial, and ovarian cancers, that is centered on *ERBB2* [11]. *ERBB* family genes are involved in the tumor biology of several solid tumors, including these three estrogen-related cancers [12]. However, the *ERBB* gene involvement in these cancers and signaling cascades are not yet fully understood [12,36]. In the present study, we applied a machine learning model on integrated miRNA omics and network data, and were able to predict common BC, UCEC, and OV deregulated miRNAs. Some of these are potentially able to target the *ERBB* family. To address these issues, we used the TCGA and miRTarBase databases to construct a weighted Support Vector Machine (SVM) learning model. We validated in vitro the most promising common miRNA signatures, using cancer and parental cell lines for each tumor type. Collectively, our results highlight common deregulated expression signatures for the estrogen-dependent cancers (BC, UCEC, and OV) analyzed here.

## 2. Materials and Methods

### 2.1. Data Collection and Machine Learning Analysis

TCGAbiolinks (v2.14.1) [37] was used to retrieve TCGA-BRCA, TCGA-OV, and TCGA-UCEC primary solid tumors and solid tissue normal HT-seq-normalized (reads-per-million miRNA) count. Clinical annotations for primary tumors were mixed histology (TCGA-BRCA), serous cystadenocarcinoma (TCGA-OV), and endometrioid endometrial adenocarcinoma (TCGA-UCEC). All data were downloaded by GDC resources on 10 May 2021.

We processed miRNA profiles of 2255 female samples, including primary tumor tissues (1096 TCGA-BRCA, 545 TCGA-UCEC, and 490 TCGA-OV)) and solid tissue normal for TCGA-BRCA (104 cases) and TCGA-UCEC (33 cases). Ovarian miRNA-seq data were not available for solid tissue normal samples (see Appendix A). From 2255 samples, we removed 13 male samples from the BRCA dataset and filtered out the vial/plate duplications using the mean as a unique value (resulting in 2229 total samples).

From a total of 1882 miRNAs, we selected all miRNAs with non-zero values for all tumor and normal samples (*n* = 1683). We retained only those miRNAs expressed as counts-per-million (CPM) above 0.5 in at least 90% of the samples (*n* = 309) and performed a global data normalization via the upper quartile method (UQUA). The final dataset was composed of 2229 (2093 tumor/136 normal) samples and 309 miRNAs. We retrieved, from miRTarBase [*Homo sapiens* (hsa ) miRNA, release 7.0] [38], the experimentally validated *ERBB2, ERBB3*, and *ERBB4* miRNA target interaction (MTI). We examined the UALCAN portal [39] for *ERBB2* and *ERBB3* expression profiling in TCGA-BRCA and TCGA-UCEC cohorts, respectively. We split the dataset into training (70%, 1562 samples) and test sets (30%, 667 samples) for the classification step. Thus, we preserved the proportion of cancer types in the data distribution (in the split data, we had the same percentage of cancer types as in the full dataset, training dataset, and test dataset). Then, we performed a feature selection using the Recursive Feature Elimination (RFE) method (caret v6.0-88), which resulted in a selection of 205 miRNAs. We selected the top 15 features using variable importance analysis and all features found in both the feature selection output and the *ERBB* miRNA target dataset from our previous study, for a total of 42 miRNAs. Given the unbalanced dataset (2093 tumor/136 normal), we performed an artificial generation of new training samples for the minority class, using the SMOTE method [40]. We used the resulting training set as an input of a Caret Support Vector Machine [41], a classification algorithm widely used in transcriptome classification studies [42,43,44] that maximizes the width of the margin between the classes and, at the same time, minimizes the empirical errors. In the training phase, we used k-fold cross-validation (k=10), a grid approach for parameter tuning (Cost, Weight, and Sigma) and a 70–30% splitting rule for the training and test set definitions.

### 2.2. CCLE Dataset Analyses, Hierarchical Clustering, and Database Explorations

We used the Cancer Cell Line Encyclopedia (CCLE) dataset (PRJNA523380) [45], including 734 miRNA profiles for 934 cell line models and annotations. We retained only CCLE IDs of female primary “breast”, “ovary”, and “endometrium” carcinoma with the following histological subtypes: “ductal_carcinoma”, “adenocarcinoma”, “endometrioid_carcinoma”, “metaplastic_carcinoma”, and “serous_carcinoma”. This resulted in a total of 58 malignant cell lines (miRNA-seq of normal samples were not available). We found 39 out of 42 miRNA features within the CCLE dataset. The clustered heatmap of miRNA profiles (log2-normalized counts) was based on Pearson correlation as the distance metric between cell lines, using the “pheatmap” R package (v1.0.12). Heatmap and clustering analysis grouped the miRNAs with similar variation in expression across cancer cell lines. In particular, we evaluated standard deviation (SD) and interquartile range (IQR) descriptive statistics for each miRNA along the cell lines. Then, we calculated quartiles of SD and IQR distributions, and subsequently selected only miRNAs with SD and IQR less than or equal to the first quartile of SD and IQR distributions. This filtering indicated nine pivotal miRNAs. Moreover, we downloaded the whole of the Human miRNA-Disease Association Database (HMDD) (version 2019.01) including tissue, circulation, genetic, and miRNA-target assay data [46] to search miRNA disease annotations. We identified 14 entries of interest for breast, ovarian, and endometrial carcinoma (“Adenocarcinoma, Endometrial”, “Endometrial Neoplasms”, “Uterine Corpus Endometrial Carcinoma”, “Ovarian Neoplasms”, “Breast Adenocarcinoma”, “Breast Ductal Carcinoma”, “Breast Neoplasms”, “Carcinoma, Breast”, “Carcinoma, Breast, Triple-Negative”, “Carcinoma, Endometrial”, “Carcinoma, Endometrioid Endometrial”, “Carcinoma, Ovarian”, “Ovarian Serous Cystadenocarcinoma”, “Early-Stage Breast Carcinoma”) associated with miRNA candidates, except for hsa-mir-337, hsa-mir-1301, hsa-mir-3127, hsa-mir-323b, and hsa-mir-1296. We used the Ingenuity Knowledge Base (version 62089861) (IPA, QIAGEN, Inc. Redwood City, CA, USA) [47] for miRNA downstream analysis, including biomarker applications.

### 2.3. Cell Culture

Breast epithelial cell line, MCF10A (ATCC^®^ CRL-10317, American Type Culture Collection 10801 University Boulevard, Manassas, VA, USA); breast cancer cell lines, MCF-7 (DSMZ # ACC 115, Inhoenstraße 7B, 38124 Braunschweig, Germany) and T47D (DSMZ # ACC 739); endometrial stromal cells, HESC (abm # T0533, Applied Biological Materials Inc. #1-3671 Viking Way Richmond, BC V6V 2J5 Canada); endometrium adenocarcinoma cell lines, MFE-280 (DSMZ # ACC 410) and EN (DSMZ # ACC 564); ovarian cell line, OCE1 (obtained from the Live Tumor Culture Core at the University of Miami Sylvester Comprehensive Cancer Center); ovary cystadenocarcinoma EFO-21 (DSMZ # ACC 235); and ovarian clear cell adenocarcinoma, ES-2 (ATCC^®^ # CRL-1978), were cultured according to manufacture recommendations. Briefly, MCF10A were cultured in MEBM^®^ (Mammary Epithelial Cell Basal Medium; Lonza, Walkersville, MD, USA) supplemented with BPE, hEGF, insulin, and hydrocortisone (MEGM^®^SingleQuots, Lonza, Walkersville, MD, USA). MCF-7 cells were grown in RPMI (Roswell Park Memorial Institute, Life Technologies, Buffalo, NY, USA) medium supplemented with 10% fetal bovine serum (FBS), with insulin (0.01 mg/mL), 2 mM L-glutamine, and 1% non-essential amino acids (neaa). T-47D were maintained in RPMI supplemented with 10% FBS, insulin (0.01 mg/mL), and 2 mM L-glutamine. HESC were grown in Prigrow IV medium, 10% FBS charcoal-stripped, and 2 mM L-glutamine. MFE-280 were grown in 45% RPMI 1640 + 45% MEM (with Earle’s salts), 10% FBS, 2 mM L-glutamine, and 1× insulin-transferrin-sodium selenite. EN was cultured in 80% Dulbecco’s MEM, 20% FBS, and 2 mM L-glutamine. OCE1 cells were cultured in Primaria cell culture flasks (BD) with FOMI Medium supplemented with 25 ng/mL Cholera Toxin. EFO-21 were cultured in 80% RPMI, 20% FBS, 2 mM L-glutamine, 1% neaa, and 1% sodium pyruvate.

### 2.4. RNA Extraction and Real-Time PCR

Total RNA was isolated from cell cultures using miRNeasy Micro Kit (Qiagen, Hilden, Germany), according to the manufacturer’s instructions, and eluted in 40 µL of RNase-free water. The RNA quantity and quality were evaluated by NanoPhotometer^®^ NP80 (IMPLEN, USA). RNA purity for assessing the protein and salt contaminants, based on the A260/280 and A260/230 ratios, was ≥1.8 for all samples. A total of 200 ng RNA was reverse transcribed using a miRCURY LNA^TM^ RT Kit (Qiagen) and used for real-time PCR experiments. Expression values of hsa-mir-381-3p, hsa-mir-33a-3p, hsa-mir-331-3p, hsa-mir-331-5p, hsa-mir-193a-5p, hsa-mir-193a-3p, hsa-mir-1301-3p, hsa-mir-1296-5p, hsa-mir-1247-3p, hsa-mir-1247-5p, hsa-mir-146b-3p, hsa-mir-146b-5p, hsa-mir-323a-3p, and hsa-mir-323b-3p were determined by real-time PCR using specific primers (Qiagen), and miRCURY LNA^TM^ SYBR Green PCR Kit (Qiagen). Using the CFX384 (Bio-Rad, Hercules, CA, USA), PCR cycling conditions were 95 °C for 2 min, 40 cycles of 95 °C for 10 s, 56 °C for 60 s, and melting curve analysis 60–95 °C. The maximum cycle threshold (Ct) value was set at 280. RNU5, GAPDH, and B2M were used as housekeeping control genes. Experiments were carried out in triplicate for each data point, and data analysis was conducted using CFX Maestro Software (Bio-Rad, USA). Data were expressed as a relative expression using the 2-ΔΔCt method (compared with normal cell lines).

## 3. Results

### 3.1. Integrated Computational Approaches 

Our previous study integrated TCGA breast, ovary, and uterine endometrial corpus gene expression profiling, to identify a top-scoring PPI network centered on the *ERBB2* gene [11]. The *ERBB* family is involved in the tumor biology of several solid tumors, including the three female hormone-related cancers [12]. The ErbB receptor tyrosine kinases exist as homodimers and heterodimers composed of *ERBB2, ERBB3,* and *ERBB4* isoforms [48]. In the present study, using a TCGA dataset, we found that *ERBB2* and *ERBB3* were overexpressed in breast and endometrial cancer tissues compared with normal counterparts (Appendix A). As the TCGA dataset lacks expression data from normal ovarian tissues, we could not compare *ERBB2* and *ERBB3* expression between cancer and normal ovarian tissues.

We attempted to identify common miRNA signatures across the three hormone-related cancers and their putative relationships with the cancer hallmark, *ERBB* isoforms. Figure 1 displays the whole workflow of the present study.

To this end, we queried two bioinformatics resources: (i) the TCGA, a genomic database, for retrieving miRNA expression profiling of large BRCA, UCEC, and OV cohorts, and (ii) miRTarBase, an experimentally validated microRNA target interactions database, for finding miRNAs targeting *ERBB* isoforms independently from tumor type. To identify miRNAs associated with the three female cancers, we leveraged a model based on the normalized counts of the miRNA.

Due to feature selection on the full dataset, our model identified 205 features (TCGA-BRCA, TCGA-UCEC, and TCGA-OV miRNA, Appendix A). On these selected features we applied a Support Vector Machine (SVM) algorithm to execute a Tumor vs. Normal binary classification. We selected this machine learning method, based on its wide usage in the field of transcriptomic and miRNA classification works, as reported in the literature [43,44,49,50,51]. Other methods (i.e., Logistic regression, Boosted Logistic Regression, Regression with LASSO penalty, Elastic Net, Random Forest, Neural Networks using Model Averaging (avNNet, and the “nnet” package)) produced similar results in AUC and other metrics, compared with SVM (the AUC obtained from the tested ML methods ranging from 0.911 to 0,97 compared to an AUC of 0.931 by SVM) as reported in Appendix A. Random Forest and avNNet performance achieved better results in terms of AUC and F1 score. However, we did not obtain deep insight into the systematic appraisal of model performance because it is beyond the scope of the study.

Using the miRTarBase, we found 158 experimentally validated miRNAs able to target *ERBB* isoforms: 91 targeting the *ERBB2* gene, 57 targeting *ERBB3,* and 10 targeting *ERBB4* (Appendix A). The intersection of the 205 features ranked according to model importance weighting, with the 158 miRNA targeting *ERBB* isoforms, resulted in 28 overlapping miRNAs from the 205 selected features. Upon testing model performance, based on different miRNA combinations, we kept the top 15 miRNAs governing model performance (Appendix A), plus the 28 miRNAs that targeted *ERBB* isoforms (i.e., 27 univocal, since hsa-mir-145 was present in both sets of miRNAs). Thereby, our joint analyses identified 42 features, ordered by importance weighting (Table 1), as potential expression signature similarity associated with the three female-specific cancers.

### 3.2. miRNA Validation in the Cancer Cell Line Encyclopedia

Our in silico approaches predicted 42 miRNAs able to discriminate tumors from normal tissues, and some of these were potentially able to target *ERBB* isoforms. To validate our in silico predictions and select the in vitro model systems for each of the three tumors, we utilized the cancer cell line encyclopedia (CCLE), a large collection of expression and genetic data for human cancer cell models [45]. The CCLE dataset has previously been used to mirror TCGA samples and cell line genomic profiling in OV cancer [52] and BC [53], respectively. However, the CCLE dataset, unlike the TCGA dataset, does not provide miRNA expression data for normal cell lines. We observed that most of the identified miRNAs (39 out of 42) were expressed across a variety of breast, ovary, and endometrium cancer cell lines (Figure 2), except for hsa-miR-139, -miR-337, and -miR-3127.

Moreover, unsupervised hierarchical clustering analysis highlighted BRCA, OV, and UCEC cell line clusters along heatmap columns. We then performed a variability data analysis, based on the standard deviation (SD) and interquartile range (IQR) for each miRNA, together with the overall breast, ovary, and endometrium cancer cell lines. In detail, we evaluated SD and IQR distributions, and selected only miRNAs with SD and IQR values less than or equal to the first quartile distribution, to capture signatures with the lowest variability in expression profile. These filtering criteria identified nine pivot miRNAs for further internal validation; these are reported in bold in Table 1.

To select miRNAs for in vitro validation, we used the following criteria: where none of the queried databases specified the miRNA isoform, we decided to analyze both the 3p and 5p isoforms; otherwise, where at least one of the queried databases established the miRNA isoform, we chose to analyze only that one (details in Appendix A). In this way, we selected 14 miRNA mature sequences, as reported in Table 2.

### 3.3. In Vitro Results

It is well-known that cancer model cell lines might be “systems” to appraise essential oncogene roles and drug responses, and, in turn, could guide precision oncology, as they are relatively easier systems than primary tumors [54,55]. Thus, to validate the in silico results (Table 2), the relative expression of miR-33a-3p, -146b-3p, -146b-5p, -193a-3p, -193a-5p, -323a-3p, -323b-3p, -331-3p, -331-5p, -381-3p, -1247-3p, -1247-5p, -1296-5p, and -1301-3p was determined in normal (MCF10A) and BC cell lines (MCF7 and T47D), in normal (HESC) and endometrial cancer cell lines (MFE-280 and EN), and in normal (OCE1) and ovarian cancer cell lines (ES2 and EFO21) (Appendix A). Among the selected miRNAs, five (miR-323a-3p, -323b-3p, -331-3p, -381-3p, and -1301-3p) showed a similar deregulated trend in all cell cancer phenotypes compared with normal phenotypes (Figure 3).

Indeed, the expression levels of miR-323a-3p, -323b-3p, and -381-3p were greatly decreased in breast, ovarian, and endometrial cancer cell lines, independent of tumor type. In the same way, miR-331-3p and mir-1301-3p expression increased in cancer cell lines compared with normal lines, although we found small but significant changes in miR-331-3p in the MCF7 vs. MCF10A cell lines, and miR-1301-3p in the EFO21 vs. OCE1 cell lines.

Hence, we wondered whether these five miRNAs could retain their expression trends in TCGA primary tumors. Due to the fact that the TCGA ovarian cohort did not include normal tissues, comparative analysis of miRNA expression levels between tumor and normal tissues was conducted only for TCGA-BRCA and TCGA-UCEC cohorts. Our results show that the TCGA findings matched those obtained by in vitro validation, except for miR-323b-3p, which was downregulated in BC cell lines, while being upregulated in BC tissues (Figure 4A), and unchanged in UCEC tissues (Figure 4B).

Taken together, these results highlight that these five miRNAs are similarly deregulated in all three estrogen-dependent cancers.

## 4. Discussion

In our previous study, we designed an integrated approach of TCGA-BRCA, -OV, and -UCEC gene expression for identifying expression signature similarity in estrogen-dependent cancers. We highlighted a leading protein–protein interaction (PPI) network underlying breast, endometrial, and ovarian cancers centered on the oncogene Erb-B2 Receptor Tyrosine Kinase 2 (*ERBB2*) [11].

*ERBB2*, also commonly referred to as *HER2*, plays an essential role in human pathobiology. Four isoforms of the ErbB family have been identified: EGFR (ErbB1, *HER1*), ErbB2 (*HER2*), ErbB3 (*HER3*), and ErbB4 (*HER4*). Regulation of *ERBB* activity depends on its homo- or hetero-dimerization. Generally, *ERBB2* is involved in mammary gland development during puberty, proliferation, and differentiation, and it is a key player in female-specific malignancies [12]. Identification of its functional pathways and networks could be essential in determining the potential *ERBB* family regulatory mechanisms in female-specific cancers. Hence, we performed an integrated bioinformatics analysis by combining in silico miRNAs discovered via a machine learning approach from TCGA-BRCA, -OV, and -UCEC datasets and the miRTarBase knowledge for in silico identification of miRNAs targeting ERBB isoforms. In the field of transcriptomic and miRNA classification, many studies adopted the SVM machine learning method because it outperformed the others [43,44,49,50,51]. We also chose an SVM approach, since other methods (i.e., logistic regression, Boosted Logistic Regression, Quantile Regression with LASSO penalty, Elastic Net, Random Forest, avNNet, and the “nnet” package) produced relatively similar results for AUC and other metrics compared with SVM. Our computational integrated pipeline predicted 42 miRNAs commonly deregulated in the three estrogen-dependent tumors, and some of these were potentially able to target ERBB family members.

To test our in silico predictions, we utilized the publicly available CCLE dataset, including breast, ovarian, and uterine endometrial corpus model systems, which allowed us to establish a set of mature miRNA sequences and a panel of cell lines. We selected similar miRNA expression patterns across the breast, ovarian, and uterine endometrial corpus cell lines. By clustering analysis and miRNA variation analysis, we prioritized a set of 14 pivotal miRNAs for in vitro validation across the breast (MCF7 and TD47), the ovarian (ES2 and EFO21), and the endometrium (MFE-280 and EN) cancer cell lines vs. parental cell lines (MCF10-A, OCE1, and HESC). Interestingly, 5 (miR-331-3p, miR-381-3p, miR-323a-3p, miR-323b-3p, and miR-1301-3p) out of 14 miRNAs showed a similar expression trend across the three tumor types compared with their parental cell lines. Furthermore, four out of five miRNAs (miR-323a-3p, miR-323b-5p, miR-331-3p, and miR-381-3p) retained their expression trend in TCGA primary tumors.

Recently, several studies investigated the miRNA regulatory mechanisms underlying estrogen-dependent cancers. In 2019, Liolios et al. [17] found eight key player miRNAs that are commonly deregulated in female reproductive system cancers. In addition, the circulating miR-765 was shown to promote endometrial cancer development via the ERβ/miR-765/PLP2/Notch axis [20]. Again, Ritter et al. demonstrated that the deregulated expression of miR-484 and miR-23a in serum and urine human samples was linked to endometrial and ovarian cancers [23]. 

The growing demand for identifying miRNAs associated with specific diseases gave rise to a common bioinformatic approach; the fusion of various public bioinformatic resources. To this end, recent studies have shown that machine learning approaches can significantly improve the analysis of next-generation sequencing expression data to find new biological patterns and genes, or groups of genes, involved in several pathologies [7]. In fact, a previous study has utilized unsupervised and supervised machine learning techniques on gene expression data with variable success rates [7]. Other studies compared the capacity of supervised methods, such as Support Vector Machines (SVM), Decision Trees, and Random Forest classifiers to perform disease/healthy sample classification tasks [31]. For instance, Ha et al. [56] proposed a novel approach for inferring novel disease–miRNA associations through a machine learning method based on matrix factorization, and used various miRNA databases, such as HMDD and miR2Disease. Machine learning is also applied to discriminate cancer subtypes. Indeed, Ali et al. [33] used a Neighborhood Component Analysis (NCA) and Long Short-Term Memory (LSTM), a type of Recurrent Neural Network, to classify a given miRNA sample into kidney cancer subtypes, using the miRNA quantitative read counts data provided by the TCGA. The results showed a subset of 35 miRNAs that could group kidney cancer miRNAs into five subtypes. The current study aimed to identify common miRNA signatures across the three estrogen-dependent cancers via a machine learning approach, and discovering hidden relationships between the *ERBB* family and its regulatory miRNAs.

Our weighted support vector machine approach, through feature selection, prioritized miRNAs already known in female malignancies, in agreement with previous findings; for instance, the eight key player miRNAs found by Liolios et al. (miR-21, miR-155, miR-145, miR-200b, miR-205, miR-34a, miR-92a, miR-101) [17], and novel miRNAs such as miR-1301 and miR-1296. This latter miRNA had low model importance upon feature selection but, importantly, was one of the miRNAs targeting ERBB isoforms, and satisfied our statistical criteria upon clustering analysis on the CCLE dataset. Indeed, recent functional findings have demonstrated miR-1296-5p involvement in the regulation of breast cancer cell progression by targeting the ERBB2/mTORC1 signaling pathway [57] and tumor suppressor roles in triple-negative breast cancer [58,59,60].

In addition, by comparing our results with a state of the art model, an ensemble feature selection strategy proposed by Lopez-Rincon et al. [61], we found that 23 out of our 42 features (about 54%) overlapped the 100 miRNA tumor signatures extracted from 28 TCGA tumor types. Interestingly, 10 out of the 23 features ranked in the top 20 features of our model importance. In agreement with the results of Lopez-Rincon et al., hsa-mir-183 and hsa-mir-10b were within our top-ranking miRNAs (#1 and #4 model rank, respectively). Consistent with features, 16 out of our 42 features overlapped the 38 clinically verified breast cancer-associated miRNAs (hsa-mir-10bhsa-mir-125b-1, hsa-mir-145hsa-mir-21, hsa-mir-125a, hsa-mir-17, hsa-mir-206, hsa-mir-155, hsa-mir-206, hsa-mir-17,hsa-mir-221, hsa-mir-200c, hsa-mir 34a, hsa-mir-146a, hsa-mir-146b, hsa-mir-205) used in the Rehman et al. study [28].

Tumor-specific miRNA expression differences have been proven in vitro to discriminate breast, ovarian, and endometrial cancer cell types [22]. Consistent with Hirschfeld et al. [22], we found 16 out of our 42 features of which mir-200c, mir-100, mir-200b were within our top 15 features for model importance. Riaz et al. used the miRNA expression profiling of 51 human breast cancer cell lines to reveal subtype and driver mutation-specific miRNAs [21]. In line with literature data, we also used model cell systems for in vitro validations, upon pivotal miRNA selection using a public cancer cell line dataset collection. Our in vitro results demonstrated that five miRNAs (miR-331-3p, miR-381-3p, miR-323a-3p, miR-323b-3p, and miR-1301-3p), had a similar expression trend across cancerous breast, ovarian, and uterine endometrial corpus cells compared with their parental cell lines. Furthermore, four out of five miRNAs (miR-323a-3p, miR-323b-5p, miR-331-3p, and miR-381-3p) retained their expression trend in TCGA primary tumors. In agreement with previously published data [62], two of these miRNAs (miR-323a-3p and miR-331-3p) would be able to target *ERBB* isoforms. miR-331 expression was deregulated in breast cancer [63], and directly targeted HER2 in breast [62], prostate [64], gastric [65], and colorectal [66] cancer cell lines. Interestingly, miR-381-3p upregulation suppresses BC progression by inhibiting epithelial–mesenchymal transition [67]. This result is in line with our data, since we found that miR-381-3p expression was strongly downregulated in BC cell lines, and in ovarian and endometrial cancer cells.

Similarly, a recent study reported that ectopic expression of miR-323a-3p inhibited the malignant behavior of BC cell lines and inhibited tumor growth [68]. In our experiments, we found that the expression level of miR-323a-3p was strongly inhibited in breast, ovarian, and endometrial cancer cell lines. We additionally found that miR-323b-3p was downregulated in all cancer cell lines. It has recently emerged that its downregulation in plasma is associated with lymph node metastases in early breast cancer patients [69]. Regarding miR-1301-3p, Peng et al. reported its downregulation in BC cell lines and tissues [70]. In contrast, our findings show an increased expression level for this miRNA in all assessed cancer cell lines, and these results agree with TCGA dataset results. Hence, our integrated analysis indicated 42 miRNA features characteristic of breast and two gynecologic tumors that are able to discriminate between tumor and normal tissues. In vitro experiments suggest that five miRNAs might be tumor signatures for the three female cancers.

Nevertheless, the present study has some limitations. The first is the lack of ML model validation on the dataset including ovarian normal counterparts, since TGCA-OV lacks these miRNA-seq data. Moreover, the data are unbalanced (2093 tumor/136 normal) and, even though we performed an artificial generation of samples through the SMOTE technique, this method does not consider neighboring values, and thus can increase the overlapping of classes and introduce additional noise [71]. Moreover, we could not find any other large public datasets of breast, ovarian, or endometrial miRNA expression profiling; therefore, we could not test the whole methodological workflow on external datasets to generalize the performance of the classifier. The third limitation is the lack of ex vivo validation, which certainly would have made the ML results more robust.

Indeed, further investigations will focus on the potential implications of the functional roles of the miRNAs and the *ERBB* family for future translational applications. One future perspective is to test whether the identified miRNAs are also deregulated in the plasma of BC, OV, and UCEC patients, as, if so, this information could be used for diagnostic and prognostic purposes.

## 5. Conclusions

In summary, this study aimed to illustrate the potential of a machine learning approach, combined with *ERBB* knowledge, to dissect similarities in miRNA expression in estrogen-sensitive female-specific tumors. Using publicly available datasets, we propose an integrated computational framework based on a binary classification model and driven by the cancer hallmark *ERBB* to discover common miRNA signatures characterizing breast, endometrial, and ovarian tumors as possible diagnostic biomarkers, which would have potential for future research in clinical practice. Indeed, further validation would be critical to ascertain the utility and specificity of these miRNAs in diagnostics. Meanwhile, from a plethora of miRNAs across three clinically distinct tumor entities, our analysis framework identified five miRNAs as potential biomarkers for the three cancer types. As a future perspective, it would be interesting to assess the miRNA regulatory network around the *ERBB* family.

## Figures and Tables

**Figure 1 biomedicines-10-01306-f001:**
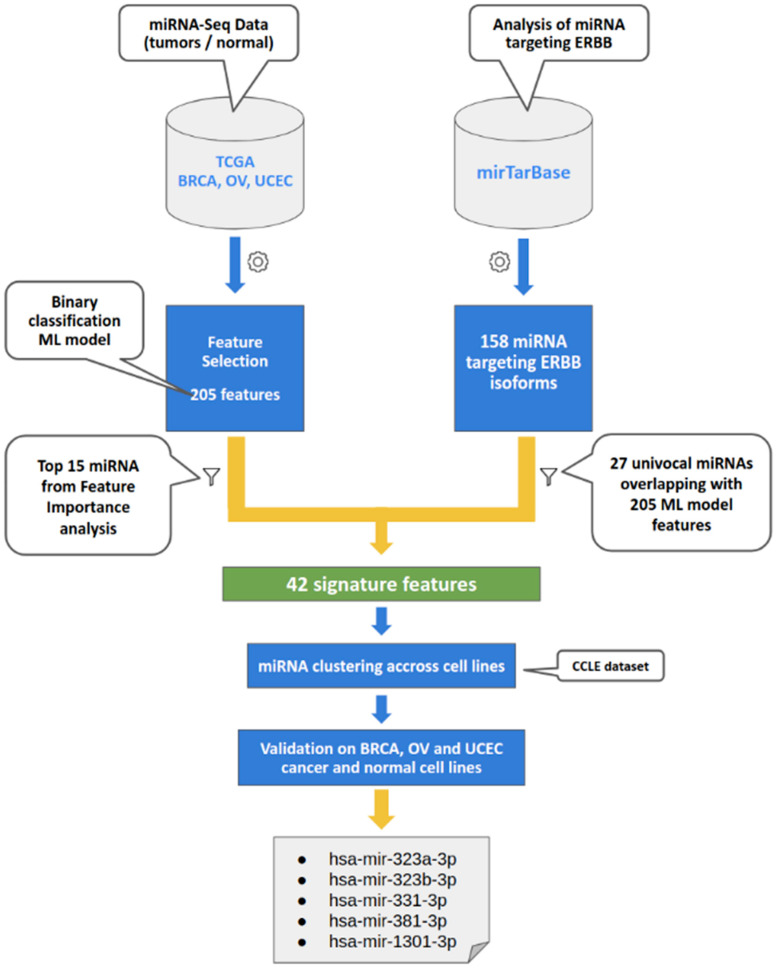
**Workflow.** The study design encompasses the integration of TCGA miRNA expression data and miRNAs targeting *ERBB* family genes within a machine learning approach to prioritize common miRNA signatures. Abbreviations: The Cancer Genome Atlas (TCGA), Breast invasive carcinoma (BRCA), Ovarian Serous Cystadenocarcinoma Cancer (OV), Uterine Corpus Endometrial Carcinoma (UCEC), miRNA Target Interaction (miRTarBase) Database, the Cancer Cell Line Encyclopedia (CCLE) dataset (PRJNA523380), miRNA expression dataset.

**Figure 2 biomedicines-10-01306-f002:**
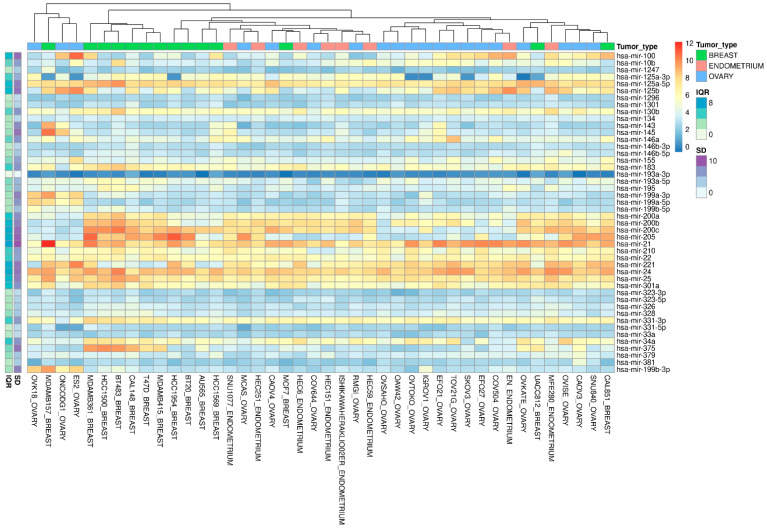
**Heatmap with hierarchical clustering of the 39 out of 42 miRNA features within the Cancer Cell Line Encyclopedia (CCLE) across female primary “breast”, “ovary”, and “endometrium” carcinoma.** Left columns show standard deviation (SD) and interquartile range (IQR) for each miRNA along all cell lines. In the columns, cell line clusters for miRNA expression signature similarity are reported.

**Figure 3 biomedicines-10-01306-f003:**
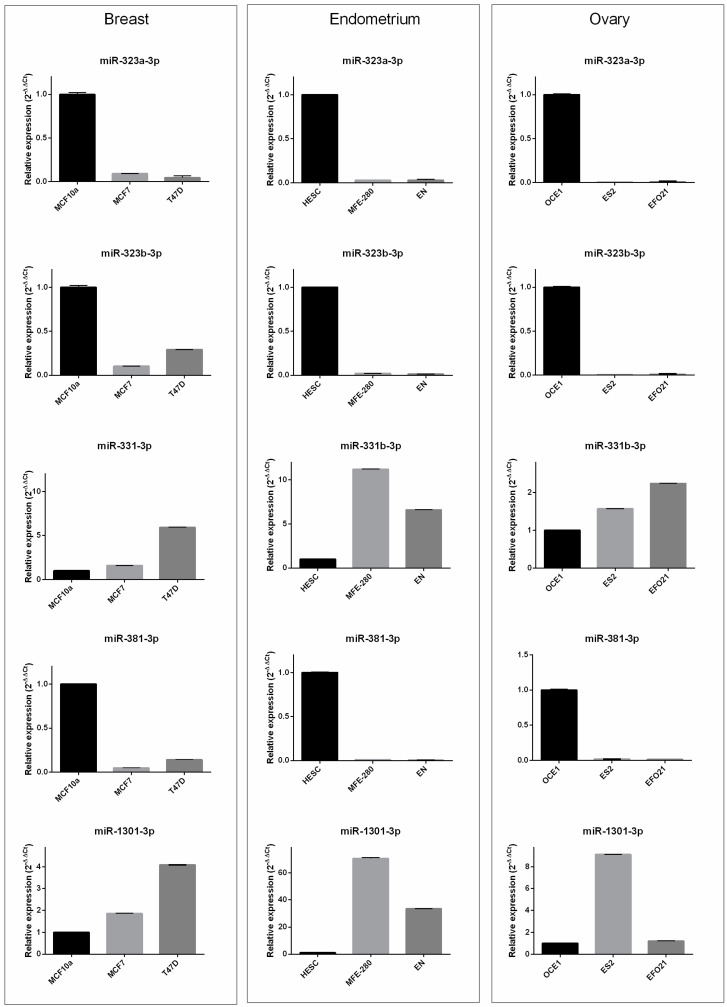
**Relative expression of selected miRNAs in cancer vs. normal cell lines.** Left panel: miRNA expression in normal breast (MCF10A) and breast cancer (MCF7 and T47D) cell lines. Middle panel: miRNA expression in normal endometrial (HESC) and endometrial cancer (MFE-280 and EN) cell lines. Right panel: miRNA expression in normal ovary (OCE1) and ovarian cancer (ES2 and EFO21) cell lines. miRNA expression was evaluated by real-time PCR (2−ΔΔCt).

**Figure 4 biomedicines-10-01306-f004:**
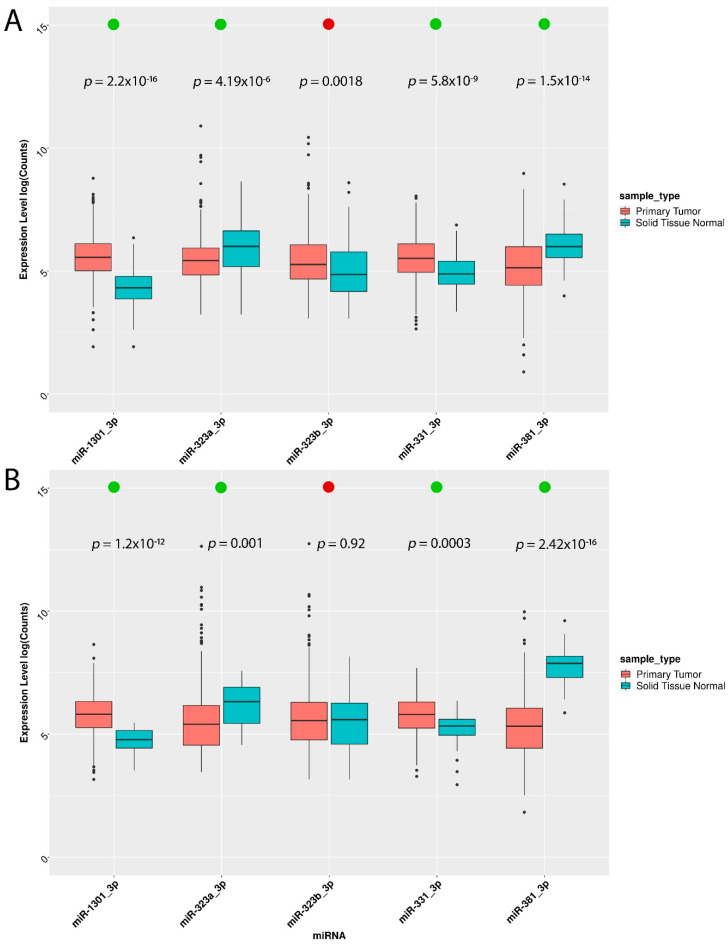
**Comparison of miRNA expression between TCGA tissues and cell lines used in vitro.** (**A**) TCGA miRNA expression (log scale) of breast invasive carcinoma (*n* = 1096) and solid tissue normal (*n* = 104) compared with relative expression in breast cell lines (MCF7 and T47D) and normal tissue cell line (MCF10A). (**B**) TCGA miRNA expression (log scale) of uterine corpus endometrial carcinoma (*n* = 545) and solid tissue normal (*n* = 33) compared with relative expression in endometrial cancer cell lines (MFE-280 and EN) and normal (HESC). The trend concordance in tissue/cell lines is indicated by the green dot, and non-concordance is indicated by the red dot. Statistically significant tumor/normal differences (*p*-value < 0.05, Wilcoxon test).

**Table 1 biomedicines-10-01306-t001:** The selected 42 features (miRNAs) associated with breast, ovarian, and uterine corpus endometrial cancers, with the corresponding feature source, and fold-change in expression (tumor vs. available normal sample normalized read counts), ranked according to the weighted support vector machine (wSVM) model. The miRNAs with ID indicated in **bold** were evaluated for in vitro validations. The importance weighting column indicates every feature’s weight.

Model Rank	miRNA ID	Importance Weighting	Feature Source ^a^	FC (BRCA)	FC (UCEC)
1	hsa-mir-183	100	Top 15	8.49	23.62
2	hsa-mir-139	93.87	Top 15	−7.03	−11.46
3	hsa-mir-145	89.75	Top 15, Targeting ERBB	−5.05	−10.80
4	hsa-mir-10b	85.99	Top 15	−3.20	−6.57
5	hsa-mir-337	84.56	Top 15	−3.72	−3.01
6	hsa-mir-200c	83.26	Top 15	3.11	3.38
7	hsa-mir-200a	82.17	Top 15	4.72	6.26
8	hsa-mir-100	81.09	Top 15	−2.93	−11.61
9	**hsa-mir-1247**	80.4	Top 15	−2.60	−11.19
10	hsa-mir-195	78.89	Top 15	−2.33	−6.27
11	hsa-mir-379	77.99	Top 15	−1.96	−4.86
12	**hsa-mir-1301**	77.47	Top 15	3.59	3.36
13	hsa-mir-210	76.05	Top 15	7.42	6.79
14	hsa-mir-200b	75.21	Top 15	3.36	4.50
15	**hsa-mir-381**	75.13	Top 15	−2.05	−7.24
16	hsa-mir-143	71.74	Targeting ERBB	−1.83	−11.42
18	hsa-mir-130b	70.99	Targeting ERBB	2.65	3.78
27	hsa-mir-3127	59.62	Targeting ERBB	2.37	1.95
30	hsa-mir-125b-1	57.64	Targeting ERBB	−3.22	−5.32
40	**hsa-mir-331**	52.59	Targeting ERBB	1.94	1.65
43	hsa-mir-134	50.36	Targeting ERBB	−1.59	−2.31
44	hsa-mir-155	50.25	Targeting ERBB	2.72	2.28
54	hsa-mir-199b	46.53	Targeting ERBB	1.01	−3.24
58	hsa-mir-199a-1	45.39	Targeting ERBB	1.10	−3.35
76	hsa-mir-22	41.32	Targeting ERBB	−1.40	−1.83
107	hsa-mir-21	32.12	Targeting ERBB	5.01	1.03
108	hsa-mir-375	31.5	Targeting ERBB	7.51	2.39
113	**hsa-mir-146b**	29.65	Targeting ERBB	1.46	1.10
114	hsa-mir-326	29.11	Targeting ERBB	−2.25	−1.14
141	hsa-mir-301a	21.2	Targeting ERBB	3.32	1.90
147	**hsa-mir-33a**	18.69	Targeting ERBB	2.67	−1.51
152	**hsa-mir-323b**	16.72	Targeting ERBB	1.19	3.21
154	**hsa-mir-193a**	16.39	Targeting ERBB	−2.02	−1.58
156	hsa-mir-205	15.28	Targeting ERBB	−2.67	51.59
157	hsa-mir-25	15.11	Targeting ERBB	−1.04	−1.01
162	hsa-mir-328	13.21	Targeting ERBB	−1.87	−1.95
168	hsa-mir-125a	10.71	Targeting ERBB	−1.31	−2.85
175	hsa-mir-221	8.38	Targeting ERBB	1.03	−3.12
180	hsa-mir-146a	7.77	Targeting ERBB	1.51	2.70
185	hsa-mir-34a	6.75	Targeting ERBB	1.21	−1.23
196	hsa-mir-24-1	2.36	Targeting ERBB	1.03	−1.54
205	**hsa-mir-1296**	0.11	Targeting ERBB	−1.44	−1.27

^a^ miRNA target interaction by miRTarBase database, *Homo sapiens* (hsa) miRNA release version 7.0.

**Table 2 biomedicines-10-01306-t002:** miRNA candidates for in vitro validations and their miRBase annotation, *Homo sapiens* (hsa) miRNA, release 21.

Mature_Acc	Mature_ID	Mature_Seq
MIMAT0005899	hsa-miR-1247-5p	ACCCGUCCCGUUCGUCCCCGGA
MIMAT0022721	hsa-miR-1247-3p	CCCCGGGAACGUCGAGACUGGAGC
MIMAT0005797	hsa-miR-1301-3p	UUGCAGCUGCCUGGGAGUGACUUC
MIMAT0000736	hsa-miR-381-3p	UAUACAAGGGCAAGCUCUCUGU
MIMAT0004700	hsa-miR-331-5p	CUAGGUAUGGUCCCAGGGAUCC
MIMAT0000760	hsa-miR-331-3p	GCCCCUGGGCCUAUCCUAGAA
MIMAT0002809	hsa-miR-146b-5p	UGAGAACUGAAUUCCAUAGGCU
MIMAT0004766	hsa-miR-146b-3p	UGCCCUGUGGACUCAGUUCUGG
MIMAT0004506	hsa-miR-33a-3p	CAAUGUUUCCACAGUGCAUCAC
MIMAT0004614	hsa-miR-193a-5p	UGGGUCUUUGCGGGCGAGAUGA
MIMAT0000459	hsa-miR-193a-3p	AACUGGCCUACAAAGUCCCAGU
MIMAT0005794	hsa-miR-1296-5p	UUAGGGCCCUGGCUCCAUCUCC
MIMAT0015050	hsa-miR-323b-3p ^a^	CCCAAUACACGGUCGACCUCUU
MIMAT0000755	hsa-miR-323a-3p ^a^	CACAUUACACGGUCGACCUCU

^a^ The closely related mature sequence hsa-miR-323a-3p, MIMAT0000755, CACAUUACACGGUCGACCUCU, was included for wet-lab experiments.

## Data Availability

The datasets analyzed in this study can be found in the TCGA database (https://portal.gdc.cancer.gov/, accessed on 10 May 2021). Details of the dataset employed in this study are provided within the Appendix A.

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
