# Peer review of "Discovering Common miRNA Signatures Underlying Female-Specific Cancers via a Machine Learning Approach Driven by the Cancer Hallmark ERBB"

_biomedicines, 2022, doi:10.3390/biomedicines10061306_

Round 1

Reviewer 1 Report

The author proposed a model to discover common miRNA signatures characterizing breast, endometrial and ovarian tumours using the SVM classifier and domain knowledge of the ERBB family. They have used a current database, namely mirTarBase. They presented a novel and effective method for miRNA study. However, I have a few suggestions.

  1. Improve “Introduction”. It would be better for the reader if the author described miRNA and how it is associated with female cancer.
  2. The author presented the model's performance is only on one split (one training set and one test set). The author split the dataset into training (70% of samples) and test sets (30% of samples) for the classification step. Therefore, the result's significance does not show (like confidence interval, standard deviation etc.). Will it be possible to split the dataset randomly (random split) into training and test five times and show the significance of the result.
  3. I would like to know the performance of the classifier on independent test dataset like the GEO dataset
  4. The author compared different classifiers; I would like to know the performance of random forest and neural network (or Deep-NN) on the dataset.
  5. Will it be possible to compare your feature set with other state-of-art models (like Lopez-Rincon, Alejandro, et al. doi: 10.1186/s12859-019-3050-8)?  

Reviewer 2 Report

Summary of the Study

In this manuscript, Pane et al. identified a set of miRNAs markers in Breast, Uterine, and Ovarian cancers targeting the ERBB family genes using a Machine learning model. They further validated them in vitro by comparing their expression in normal and cancer cell lines. Although the manuscript is well written, the authors need to explain/change a few things to improve it.

Comments to Authors:

  • Authors should use F1 scores instead of just accuracy for the performance of the ML model. Also, for which curve are the authors calculating and showing AUC?

  • Figure S1: Put the p-value on boxplots (t-test) to show a significant difference.

  • Have you tried using Random Forest ML algorithm on this data? It is a widely used and better-performing classification model for the expression data.

  • For calculating the fold-change, have you normalized the read count? Also, it seems like the fold change for the first gene in BRCA is 8.49 instead of 8,49?

  • Figure 2: Variance is shown in the legend but not anywhere else in the figure?

  • The selection of the miRNAs for in vitro validation is not good. For example, according to the ML model, the miRNA hsa-mir-1296 had very low model importance, which means it is not effective in classifying cancer and normal samples and possibly be a housekeeping gene. This means that it will have low variance in the CCLE data (your criteria for selecting final miRNAs) as there is no normal. Filtering the miRNAs based on some model importance threshold will make the pipeline better.

  • Authors should show p-values for figures 3 (carry out t-test and show p-values) and 4 (write p-values) to emphasize the significance.

  • In the discussion/introduction, the authors need to search more thoroughly about previous research. On a quick search, I found this paper (https://www.ncbi.nlm.nih.gov/pmc/articles/PMC6468888/) that identifies miRNAs markers using an ML model in breast cancer. Authors should search more about previous research and compare their results with them.

Round 2

Reviewer 2 Report

I thank the author for updating the manuscript. The updated manuscript is looking good, and the authors have addressed all of my concerns.